# Who Are the Intended Users of CSR Reports? Insights from a Data-Driven Approach

**Charlie Lindgren *** , **Asif M. Huq** and **Kenneth Carling**

School of Technology and Business Studies, Dalarna University, Högskolegatan 2, 791 31 Dalarna, Sweden; ashu@du.se (A.M.H.); kca@du.se (K.C.)
* Correspondence: clg@du.se

**Abstract:** There is extant research on theorization, conceptualization, determinants, and consequences of corporate social responsibility (CSR). However, what firms include in their CSR or sustainability reports are much less covered and are predominantly covered in case studies of individual firms. In this paper, we instead take a holistic view and simultaneously explore what firms around the globe currently disclose in these reports, more specifically we investigate if firms are shareholder or stakeholder focused. In this investigation, we check the alignment of the reports to the materiality framework of Sustainability Accounting Standards Board (SASB) which was developed having shareholders as the intended user. To estimate what firms disclose in CSR reports we used the unsupervised Bayesian machine learning approach latent Dirichlet allocation (LDA) developed by Blei et al. We conclude that firms target shareholders as the intended users of these reports, even in environments where stakeholder approach of management is argued to be more dominant. Methodologically, we contribute by demonstrating that topic modeling can enhance the objectivity in reviewing CSR-reports.

**Keywords:** CSR; sustainability; text mining; topic modeling; big data

**JEL Classification:** C8; M14

## 1. Introduction

Corporate disclosures remain an important topic for academic researchers and practitioners alike. In recent years voluntary non-financial disclosures, such as corporate social responsibility (CSR) or sustainability reports have received much attention. The terms 'CSR' and 'sustainability' are often used interchangeably in the literature though they encompass slightly different things. The term CSR is more frequently used and is more normative in nature. It refers to environmental, social, and governance-related issues and is largely considered voluntary [1–4]. Throughout the paper, we use CSR to refer to notions, activities, policies, and intentions that firms undertake for their impact on society and environment.

Firms' need for reporting [5], disclosure level [6], frequency [7], and verification of CSR disclosures [8] has increased significantly over the decades. There is extant research on theorization, conceptualization, determinants, and consequences of CSR. For example, we know that CSR is argued to be used: to portray stakeholder orientation, as a legitimizing effort, to signal asymmetric information, and to institutionalize firms [3,9,10]. It has multiple dimensions and discourses [11,12], and often influenced by firm attributes, societal norms, culture, and institutional environment [2,3,13–16]. However, what and for whom firms disclose information in their CSR reports are less covered [2,17–19], ever-changing [20], and requires more comprehensive content analysis [21].

Our paper aims to determine the intended users of firms' CSR disclosures by a comprehensive content analysis. There is a debate about who the intended users should be for these disclosures and what should be disclosed [22,23]. For example, among two

of the world's largest CSR reporting frameworks, Sustainability Accounting Standards Board (SASB) adopts an 'reasonable investors' perspective when designing their reporting framework (i.e., what information should be considered *material*), while Global Reporting Initiative (GRI) opts for a much wider spectrum of users – the stakeholders. However, we have limited empirical evidence on who the firms' view the intended users as for these disclosures in terms of 'materiality' determination [24]. In this paper, we use the materiality definition adopted by SASB. The SASB adopts the U.S. Supreme Court's definition of materiality. According to the U.S. Supreme Court, "an information is material if there is a substantial likelihood that the omitted or misstated item would have been viewed by a reasonable resource provider as having significantly altered the total mix of information". The materiality map is available at https://materiality.sasb.org/. In this paper, we therefore provide additional evidence on who the firms account to and what they account for. It is important to study CSR disclosures as it has implications for preparers, users, regulators, standard setters, and as well as the educators [23]. It bears economic consequences for the firms thus directly affect preparers and users while standard setters (e.g., SASB and GRI) and regulators use valuable resources to design various reporting guidelines.

For example, better linguistic quality in reporting leads to higher firm value [25], higher charitable contributions lead to greater revenue growth [26], better CSR scores leads to lower cost of equity [27], and bad CSR scores leads to higher interest rates on bank loan [28]. It has been also found that sin stocks (stocks of firms producing alcohol, tobacco, and gaming) are less preferred by some institutional investors and often have high litigation risk [29], and they also face greater scrutiny in their CSR communication [30].

To determine the intended users, we have collected CSR reports of some 9500 firms pertaining to the years 1998–2017 from the GRI database. These reports were subsetted and subjected to topic modeling to estimate the topics reported up on. In our devised testing strategy, the alignment of the reported topics to the anticipated topics under a shareholder perspective retrieved from the SASB are tested.

The remainder of the paper is organized as follows. The conceptual framework and the research question are given in Section 2. In Section 3, we describe the data and outline the topic modeling method as well as the testing strategy for the research question. Section 4 gives the results and Section 5 concludes with a summary and a discussion of the implication of the results.

## 2. Conceptual Framework

Determination of materiality is largely subjective and relies on the judgement of standard setters and preparers since it is determined ex-ante by them from the users' perspective [31–34]. From a standard setter's perspective, users of corporate disclosure are argued to be determined by two dominant perspectives or approaches, the 'reasonable investor' or the 'stakeholder' approach. Shareholders are predominantly considered to be the users of financial disclosure in determining what information would be considered material for decision usefulness of the users [35–38]. Reasonable investors or financiers primarily consist of shareholders, bond holders, creditors, and other lenders. Throughout the paper we use the term "shareholder" to denote reasonable investor. We acknowledge that there is fundamental difference between these groups in operational terms. In the non-financial disclosure, however, some standard setters also take a broader view on users and use stakeholder approach. According to SASB, " . . . standards focus on financially material issues because our mission is to help businesses around the world report on the sustainability topics that matter most to their investors." The standard setting process is explained in Appendix B. SASB's materiality map, given in Figure 1, identifies the material sustainability issues for shareholders in different sectors, while the content and specifics of reporting on a particular sustainability issue identified as material for multiple sectors may vary. On the other hand, GRI adopts a multi-stakeholder approach [23], " . . . GRI Sustainability Reporting Standards (GRI Standards) are created and improved using a consensus-seeking approach, and considering the widest

possible range of stakeholder interests, which includes business, civil society, labor, accountancy, shareholders, academics, governments and sustainability reporting practitioners." (https://www.globalreporting.org/information/news-and-press-center/Pages/Beyond-the-financials-Why-sustainability-reporting-needs-to-be-multi-stakeholder-in-its-approach.aspx).

| Dimension | Sub-Dimension (General Issue Category) | Consumer Goods | Extractives & Minerals Processing | Food & Beverage | Health Care | Infrastructure | Resource Transformation | Technology & Communications | Transportation |
|---|---|---|---|---|---|---|---|---|---|
| Environment | GHG Emissions | | | | | | | | |
| | Air Quality | | | | | | | | |
| | Energy Management | | | | | | | | |
| | Water & Wastewater Management | | | | | | | | |
| | Waste & Hazardous Materials Management | | | | | | | | |
| | Ecological Impacts | | | | | | | | |
| Social Capital | Human Rights & Community Relations | | | | | | | | |
| | Customer Privacy | | | | | | | | |
| | Data Security | | | | | | | | |
| | Access & Affordability | | | | | | | | |
| | Product Quality & Safety | | | | | | | | |
| | Customer Welfare | | | | | | | | |
| | Selling Practices & Product Labeling | | | | | | | | |
| Human Capital | Labor Practices | | | | | | | | |
| | Employee Health & Safety | | | | | | | | |
| | Employee Engagement, Diversity & Inclusion | | | | | | | | |
| Business Model & Innovation | Product Design & Lifecycle Management | | | | | | | | |
| | Business Model Resilience | | | | | | | | |
| | Supply Chain Management | | | | | | | | |
| | Materials Sourcing & Efficiency | | | | | | | | |
| | Physical Impacts of Climate Change | | | | | | | | |
| Leadership & Governance | Business Ethics | | | | | | | | |
| | Competitive Behavior | | | | | | | | |
| | Management of the Legal & Regulatory Environment | | | | | | | | |
| | Critical Incident Risk Management | | | | | | | | |
| | Systemic Risk Management | | | | | | | | |

**Figure 1.** SASB map depicting the materiality by dimension, sub-dimension, and sector, where dark grey is the most material. The map given in Figure 1 is given in reduced form to only include the eight chosen sectors for this analysis, the complete interactive materiality map is available at https://www.sasb.org/standards-overview/materiality-map/.

Similarly, firms may either resort to shareholder approach [39] or stakeholder approach [40] in deciding, managing, performing, or reporting CSR activities. However, it is extremely difficult to determine materiality in CSR information given its of interest to multiple stakeholders and different CSR topics are of varying importance to different stakeholders. Moreover, firms allegedly use CSR to legitimize their position in the society [3,13,41,42]. Therefore, a deviation from a shareholder perspective to stakeholder perspective can possibly also be legitimizing efforts by firm. It is also argued that in managers' decision-making process of disclosure choice materiality is often trumped by strategic choices [41]. Strategic responses may be a result of changes in internal or external environments [43]. For example, firms in the consumer goods sector may be forced to address sustainability issues such as greenhouse gas emissions due to political or social pressure even though it is not materially significant for their shareholders. While firms in financial sector may be forced to address labor practices and employee health and safety due to employee pressure [44] even though it is not materially significant for their shareholders. Firms' business environment [45,46], environmental impact [20,47–49], size [6,10,49–52], and stakeholders' influence [53,54] have also been shown to influence disclosure behavior. Thus, it is not clear who are the intended users of CSR disclosures and lead to our primary research question: Who are the intended users for disclosure of CSR information?

Given SASB's determination of material sustainability issues, hereafter referred to as dimensions, is focused on shareholders, we can expect firms' disclosure intensity across the various CSR dimensions to overlap that of SASB if firms indeed take a shareholder approach. Specifically, firms should have high intensity disclosures for a particular CSR



dimension considered material by the SASB standards in specific sectors compared to other CSR dimensions while these other CSR dimensions may have little or no disclosure. Several studies have exploited this difference that SASB distinguishes materiality based on shareholder approach and have shown that investors price CSR dimensions that are classified material by SASB. For example, stock prices of firms that reports on SASB identified material dimensions have greater stock price informativeness [55] and outperforms [56] those that do not. Moreover, none of these studies found any such significant relation for SASB immaterial dimensions disclosed in these reports even though the studies were carried out before the SASB guidelines were published. Taken together, these results indicate that investors regard the materiality issues identified by SASB important and the immaterial issues not so relevant. These finding are not surprising since value relevance for shareholders has been one of the important aspects in SASB's materiality determination process.

On the other hand, if firms commit to use a particular reporting framework which takes a multi-stakeholder approach (e.g., GRI) then one can argue firms' disclosure choice is not solely motivated by shareholders' decision usefulness. Therefore, for firms who use the GRI reporting framework we can expect a deviation in reporting from the sustainability dimensions identified by SASB as material and thereby reporting on a wider set of sustainability dimensions. It is argued that firms often use GRI frameworks to create an image or adopt a label [8] but it is also true that GRI users are more accountable [15] and credible [57] in reporting (c.f. Proposition 1 in Table 1 in Section 3.3).

Firms' reporting choices are also influenced by the business environment in which they operate [45,46]. Business environments are a function of cultural systems, financial systems, legal systems, and norms [14]. Some business environments promote stakeholder orientation while others promote shareholder orientation. It has been shown that legal environments are a good proxy to discriminate between these two dominant business environments [58–61] and the extent of social and environmental responsibility firms take [62,63]. Countries of common law legal system origin are argued to have a strong shareholder orientation while countries of civil law legal system origin have a stronger stakeholder orientation [23,58–61] (c.f. Proposition 2 in Table 1 in Section 3.3).

These institutional differences can also be observed in the accounting standard setting processes. In common law countries, accounting standards evolve from practice while in the civil law countries the State usually prescribes the detailed procedure of reporting. Which is why countries with common law legal systems are argued to have a market focus while countries with civil law legal systems to have a policy implementation focus [58,61]. Moreover, EU, where the greatest number of countries have the civil law legal system, have had the most institutional intervention (mostly voluntary) in the CSR paradigm compared to the rest of the world [8]. Firms usually adopt to institutional environment to sustain and grow [13,45]. The distinction in orientation can also be noticed between the two largest standard setters of CSR disclosures, SASB being a North American establishment and having a shareholder focus, and GRI being a European establishment and having a multi-stakeholder focus, when designing reporting standards framework. With these arguments taken together, we would expect firms in civil law legal environment to report on a wider range of sustainability dimensions within sectors than those identified material by SASB compared to firms in common law legal environment.

**Table 1.** Decision criteria

| Step 1 | Step 2 | Step 3 | Step 4 |
|---|---|---|---|
| **Model Fitting and Validation** | **Model Expansion and Investigation of RQ** | **Augmented Testing of RQ by Exploiting Proposition 1** | **Verification of RQ by Exploiting Proposition 2** |
| **5-topic model** will be first run at aggregate level to validate the fitting of latent topics through the unsupervised Bayesian model. It will then be run at a sectoral level under the assumption that firms take shareholder approach, and if so, then the distribution of the latent topics estimated by the unsupervised Bayesian model for the eight sectors will overlap with SASB materiality map for the eight sectors. | **26-topic model** is estimated under the assumption that the properties of 5-topic model and 26-topic model are comparable given the 26 topics are derived from the 5 high-level topics. Except that there may be some overlaps in the topics given the topic boundaries are not always clearly defined in the narratives of the reports. | **26-topic model** is estimated for two groups of firms: One group that uses the GRI reporting framework and the second group that do not. Based on past literature and GRI's declaration, the framework assumes a stakeholder orientation. | **26-topic model** is estimated for two groups of firms: One group that includes firms headquartered or operating in common law legal environment and the second group comprising of firms operating in civil law legal environment. Theory predicts business environment in common law legal environment is more attuned for shareholder orientation while civil law legal environment is more attuned for stakeholder orientation. |
| If the model can be validated at the aggregate level, then the following possible conclusions can be drawn from the sectoral level estimation. Shareholder orientation conclusion—if the distribution of the topics overlaps with SASB materiality map. Alternative conclusion—stakeholder orientation or prevalence of legitimization effort–if the distribution of the topics does not overlap with SASB materiality map. Model invalid—if no apparent CSR related topics are estimated due to for example noise, incorrect model fitting or data quality issues. | There are three alternative conclusions: Alternative 1: Topic distribution of the 26-topic model estimation closely follow the SASB materiality map—i.e., close to 26 distinct topics with the similar distribution as SASB predicts are identified. In such a scenario we will conclude firms adopt shareholder orientation. Alternative 2: Topic distribution of the 26-topic model estimation are more dispersed/diffused compared with the SASB materiality map—i.e., there are close to 26 distinct topics but are smeared across the matrix and not concentrated. In such a scenario, we will conclude firms adopt stakeholder orientation. Alternative 3: Results are inconclusive—i.e., no clear indication of inclination towards either approach. | There are two alternative conclusions: Alternative 1: If indeed GRI reporters adopts a stakeholder orientation by adhering to the GRI framework we will observe close to 26 distinct topics but smeared across the matrix and not concentrated. Alternative 2: If GRI reporters do not adopt a stakeholder approach then we may observe close to 26 distinct topics with similar distribution as SASB predicts. | There are two alternative conclusions: Alternative 1: If the differences in business environment influence the reporting choice of intended users then for firms in common law legal environment, we will observe close to 26 distinct topics with similar distribution as SASB predicts. On the other hand, in a civil law legal environment we may observe close to 26 distinct topics but smeared across the matrix and not concentrated. Alternative 2: If the differences in business environment do not influence the reporting choice of intended users then we will not observe any difference in distribution of the topics across the two groups. |

### 3. Data and Method

*3.1. CSR Reports*

As part of a research project at the Microdata Analysis department of Dalarna University, the starting point was to collect all available CSR reports through web scraping. For practical reasons, as a starting point, the GRI database was chosen to acquire CSR reports. The database centrally holds sustainability disclosures of some 9500 firms and has 47,093 aggregate data points from 1998 to early 2017 in multiple languages. In this study, we only analyze the reports reported in English language since it is the most common language of international business, and it alleviates expected errors from translation [52]. The GRI database also contains integrated reports, we choose to exclude these because of our focus on CSR related topics by using the firm submitted indicator variable. To be able to include all CSR reports of the GRI database in our study we would have to pre-process the integrated reports to clear the financial content, which is out of the scope of this study. Finally, 671 CSR reports as of 2016 from 61 countries were analyzed. Detailed data collection and pre-processing is explained in Appendix A.

*3.2. Topic Modeling Method*

Natural language processing is one of the most complex problem in artificial intelligence and computer science, which gives rise to several issues one needs to reckon with when analyzing text data [64,65]. For instance, one needs to consider the distinction between lexical level and semantical level, words with more than one possible meaning, synonymous words, and high frequencies of "meaningless" words, primarily stop words (such as conjunctions, prepositions, etc.) [64,66,67]. Some of these issues are dealt with during the modeling phase, while others during the data pre-processing phase. It is apparent that firm disclosures are not randomly occurring words, rather carefully constructed strategic communication. Therefore, these communications deemed to have some underlying structures and patterns. In simple terms, through complex estimation, various topic-modeling algorithms ascertains these structures and patterns. Firm disclosures are a collection of words occurring in some context, while there are common words that occur in multiple contexts. Topic modeling identifies the pattern in which these collections of words occur in some context, where the context (in this case the sustainability dimensions) is essentially a topic [64]. In the case of CSR reports these topics are essentially the sustainability dimensions or sub-dimensions that firms address in these reports.

We employ a topic modeling method on the pre-processed data consisting of document-term matrices. For analysis such as content analysis, in supervised learning, the researcher(s) beforehand sets the classification for labeling the data, which is more commonly known as the outcome variable. On the other hand, in unsupervised learning, the outcome variable is not pre-defined, rather after the analysis of the data the model gives us the possible unlabeled outcome variables based on patterns observed in the data [68]. Thus, in our case, the unsupervised learning will tell us what the dominant topics are reported in the underlying data (i.e., the CSR or sustainability reports) instead of us setting the expected topics beforehand and employing a supervised learning method to classify the data according to those topics. Through this approach, the outcomes are explicitly captured from the underlying data thereby minimizing the chances of introducing researchers' bias and inadvertently defining a set of topics that is only a subset of topics in the given data. There are quite a few methods to choose from, and each has its own advantages and disadvantages. One of the earliest probabilistic models was developed by Hofmann [64], known as the probabilistic latent semantic analysis (PLSA). While it is was a major step forward of topic modeling, one shortcoming was that it did not have a probabilistic model at the document level, but only at topic and word level [69]. This shortcoming was later addressed by Blei et al. [1] in their three-level hierarchical Bayesian method, the latent Dirichlet allocation (LDA). LDA has several extensions, one principal method being correlated topic model (CTM) developed by Blei and Lafferty [70]. The main difference between CTM and LDA is that CTM allows for correlation between topics while LDA does not. However, this comes

at some cost: CTM estimates can be biased compared to LDA [70], may contain too many general words in topics, and requires more complex computation [69].

We use the unsupervised hierarchical Bayesian machine learning approach LDA in our analysis. LDA is principally like k-means cluster analysis where the practitioner needs to decide a priori the number of latent topics, $T$ say, or number of clusters. It employs a generative probabilistic model on the collection of discrete data (i.e., each document, in our case each report), formally called text corpus (i.e., the collection of document-term matrices) to identify the underlying latent topics for a mixture of words. In doing so, it assumes: (i) every document is a mixture of a finite number of latent topics, and that these topics are distributed in different proportions in each document; and (ii) every topic is a mixture of words which may occur in different frequencies across these topics, and words are not mutually exclusive between different latent topics. The assumptions are thus minimal and appropriate to analyze reports such as sustainability or CSR disclosures. More details on the LDA model are given in Appendix C.

A distinguishing feature of LDA analysis from k-means clustering, though, is that k-means produce disjoint clusters while LDA can characterize documents by one or more topics. There has been a range of 3 to 11 topics in previous studies on CSR disclosures [20,42,51,71–75]. The studies have incorporated inductive approaches to deciding the number of topics, such as reasoning within the research group which topics are most prevalent [20].

We, however, set a priori the topics to 5 and 26 topics, to align with the 5 dimensions and 26 sub-dimensions of the SASB materiality map in Figure 1. The analysis was initially conducted on uni- or bigram document-term matrices (i.e., constructs consisting of one word and two words, respectively) with and without stemming of words where stemming refers to extracting only the stem of a word. An example is 'regul' which contains regulating, regulation, and regulations, among others. It was found that stemmed bigrams were the most interpretable as these revealed the most cohesiveness within the topics. Unigrams tend towards company specific topics or industry related terms, like practicalities within the healthcare sector such as medical procedures or mining practices within extractive minerals and processing, making topic modeling nonsensical. Stem completion was not conducted as the bigrams were easily interpretable in of themselves. In the implementation of LDA, there are two hyper-parameters to be set which are the 'document-topic prior', $\alpha$, and 'topic-word prior', $\beta$. To achieve a sparse word and topic distributions they should be set to values below 1. In the pre-processing stage, we elaborated with several settings of $\alpha$ and $\beta$ and found settings as 0.1 and $1 \times 10^{-6}$ to be sensible. In particular $\beta$ had to be set low to favor the inclusion of sparse bigrams to achieve distinct clusters. These hyper-parameters were thereafter kept constant throughout the analysis. We use the package "text2vec" [76] in R to implement the LDA analysis, which has the distinct advantage of having a $O(1)$ time complexity using the state-of-the-art sampling algorithm WarpLDA developed by Chen et al. [77].

### 3.3. Strategy to Test the Research Question

Turning to the stated research question in the conceptual framework given in Section 2, we propose a strategy to test it empirically. As a point of departure, we purport that a shareholder perspective is implemented in reports rather than a broader stakeholder perspective and investigate whether this claim holds true. Topic modeling is inherently difficult to combine with formal statistical inference, whereby subjective analysis is commonplace. We propose, however, a novel approach to testing the research questions with the scope of enhancing the objectivity of the analysis. Firstly, we convert the sub-dimensions of the SASB materiality matrix in Figure 1 to the integers 1, 2, and 3 where 1 corresponds to sub-dimensions not likely material to the sector (white), 2 corresponds to sub-dimensions material for fewer than 50% of industries in a sector (grey), and 3 corresponds to sub-dimensions material for more than 50% of industries in a sector (dark grey). To fix the score for a dimension, the average of its sub-dimensions is computed sector by sector.

Secondly, we perform the topic modeling to obtain the data-driven topics from the reports. For each sector, the output of the topic modeling provides a proportion for each topic and a low, medium, or high topic proportion is converted in to 1, 2, and 3, respectively. Assuming the topic modeling performed perfectly captures the distribution of sustainability dimensions across sectors and that firms adopt a shareholder perspective in preparing these reports, we would expect perfect concordance with the distributions retrieved from the SASB materiality map and sectorial topic distribution. Finally, the concordance of these two distributions is checked by computing the correlation and the corresponding *p*-value using the *t*-test of association between paired samples.

The expected, perfect concordance of the two distributions is arguably an idealization. We do not believe the topic modeling method to perfectly identify the latent topics and their importance in the reports. Even if the claim of shareholder perspective in reporting is true, we do not expect the reports to align perfectly with the SASB dimensions and sub-dimensions. We recognize that representing the sectorial dimension and topic proportion distributions by three levels is but an approximation. Hence, observing some discordance of the two distributions should not be a basis for refuting the claim. Such discordance could arise due to the estimation imprecision in spite of a valid claim.

To complement the procedure above, we device a multi-step testing strategy outlined in Table 1 to be able to systematically decide whether the purported answer to the research question is corroborated or refuted. In the first step, the topic modeling is checked for validity by comparing the two distributions for the five dimensions and check the most occurring bigrams per topic as to whether they fit the content of the dimensions. Upon concluding that the topic modeling is valid, the concordance of the sub-dimensions and a 26-topic model is computed in the second step. High concordance would imply corroboration and discordance refutation, whereas some discordance implies the test being inconclusive. In the inconclusive case, we proceed to the third step where concordance is compared between reports in GRI and non-GRI frameworks. The conjecture is that the framework of GRI holds a more stakeholder-oriented perspective which should materialize in greater discordance. Such observed discrepancy implies refutation, and vice-versa. The fourth step is a verification of corroboration/refutation in step 3, by comparing the discrepancy of concordance between reporting in civil law and in common law origins. Firms of civil law origins are presumed to report more stakeholder oriented. Hence, corroboration/refutation can be achieved in the second step. If not, corroboration/refutation can be achieved in the fourth step up on verification whereas non-verification in the fourth step implies an inconclusive result.

## 4. Results

The results are presented in accordance with the stepwise testing strategy outlined in Section 3.3. Figure 2 depicts the concordance between the five dimensions of the SASB materiality matrix and the data-driven topics obtained from topic modeling. The greater the (positive) correlation between a dimension and a topic, the greater the circle in the Figure 2. In the circles the *p*-values are provided where it should be noted that a small *p*-value implies a large positive correlation. White cells in the matrix indicates no connection between the dimension and the topic, or even being in opposition to each other.

Evidently, the dimensions 'human capital' and 'business model' & 'innovations' are identified as topics 2 and 4, whereas 'leadership' & 'governance' are matched by topic 5, albeit less evidently. The topic modeling for social capital was less successful with topic 3 being the only, yet weakly, positively correlated topic. As the remaining dimension and topic, one might connect 'environment' with topic 1. Such a connection would be a stretch, however: the correlation is weak and insignificant, and the topic is actually connected more strongly to 'human capital'. 'Environment', as a construct, is somewhat overarching and may easily overlap with other topics as well as dimensions. In fact, the dimensions 'business model and innovation', 'leadership and governance', and 'human capital' are as

constructs neighbors and perhaps not surprisingly somewhat difficult to disentangle in the topic modeling.

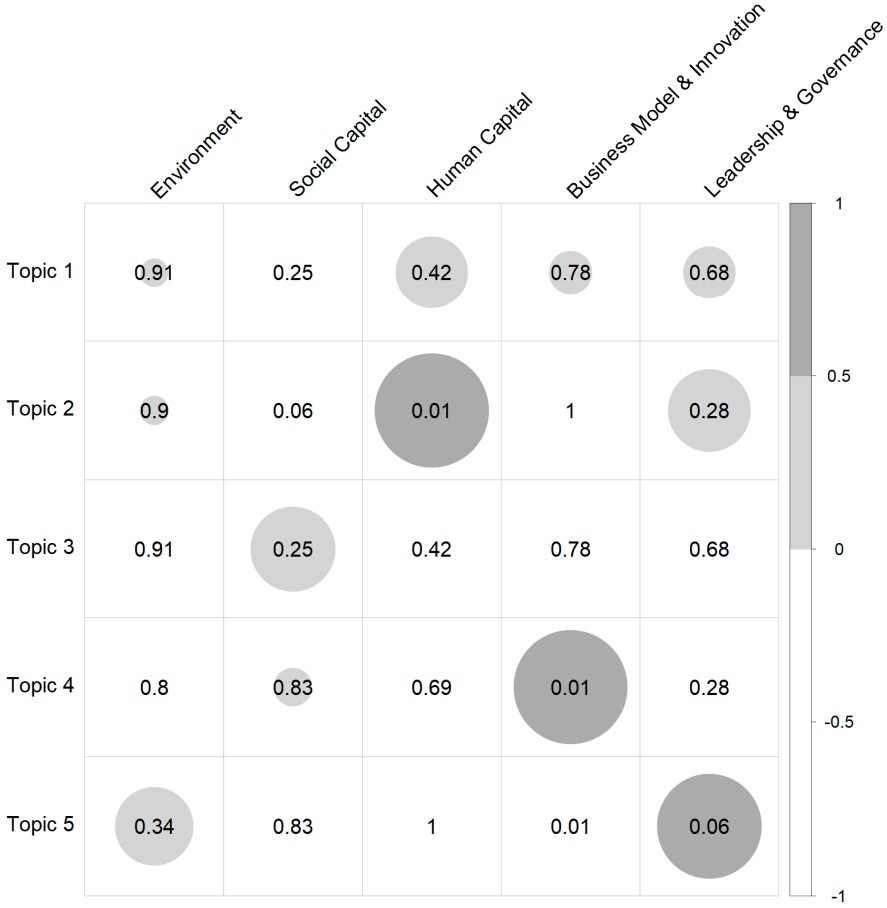

**Figure 2.** Correlation matrix for the five dimensions and the data-driven topics with *p*-values.

The topic modeling was done in R with the package "text2vec". A useful accompanying package is "LDAvis" which offers a web-based interactive visualization of topic estimation for LDA [78]. "LDAvis" is a tool to answer three main questions: (1) What is the meaning of each topic?; (2) How prevalent is each topic?; (3) How are the topics related to each other? We utilize this tool to examine the f topics. On the one hand, we note that topics are well separated and the setting of the hyperparameters sensible. On the other hand, we extract the predominant bigrams per topic and list them in Table 2. For instance, topic 2 was highly correlated with the dimension Human Capital and we therefore expect the corresponding bigrams in Table 2 to pertain to human capital. Under this topic we note stemmed bigrams like "social respons", "human resourc", and "develop employe". There is a parameter, denoted $\lambda$, that allows for identifying bigrams unique to a topic. $\lambda$ set to one is default, ranking bigrams based on frequency alone, whereas a lower $\lambda$ downplays non-unique bigrams. The table provides the bigrams for $\lambda$ set to one and to 0.3, respectively. Cross-checking the bigrams for a topic and the indicated dimension reveals a generally good match between the topics and the dimensions. Hence, and referring to the stepwise strategy, we conclude that the topic modeling, although imperfect, is valid in the sense that it has estimated topics associated with the dimensions.

**Table 2.** Selected SASB materiality dimension according to correlation matrix results, with corresponding bigrams for $\lambda = 1$ and 0.3

| $\lambda = 1$ Dimension | Topic | Bigram 1 | Bigram 2 | Bigram 3 | Bigram 4 | Bigram 5 | Bigram 6 | Bigram 7 | Bigram 8 | Bigram 9 | Bigram 10 |
|---|---|---|---|---|---|---|---|---|---|---|---|
| Environment | 1 | human right | suppli chain | health safeti | greenhous gas | climat chang | product servic | energi consumpt | local communiti | code conduct | ghg emiss |
| Human Capital | 2 | social respons | human resourc | sustain develop | corpor social | oil gas | electr vehicl | develop employe | electr power | oper perform | order ensur |
| Social Capital | 3 | corpor govern | sustain develop | research develop | corpor social | code ethic | control system | intellectu properti | year old | translat english | refer air |
| Business Modeland Innovation | 4 | corpor respons | raw materi | around world | best practic | carbon footprint | reduc environment | metric ton | young peopl | team member | rais awar |
| Leadership and Governance | 5 | fair valu | intern control | corpor govern | statutori auditor | chief offic | third parti | vote right | commerci code | take account | real estat |
| $\lambda = 0.3$ Dimension | Topic | Bigram 1 | Bigram 2 | Bigram 3 | Bigram 4 | Bigram 5 | Bigram 6 | Bigram 7 | Bigram 8 | Bigram 9 | Bigram 10 |
| Environment | 1 | human right | health safeti | suppli chain | greenhous gas | climat chang | product servic | local communiti | code conduct | energi consumpt | ghg emiss |
| Human Capital | 2 | electr vehicl | develop employe | oper perform | person data | employe hire | system oper | custom expect | natur environ | program provid | develop technolog |
| Social Capital | 3 | research develop | code ethic | intellectu properti | translat english | refer air | control system | pro tabil | intern regul | product capac | properti right |
| Business Model and Innovation | 4 | corpor respons | around world | carbon footprint | reduc environment | metric ton | team member | rais awar | young peopl | continu work | environment issu |
| Leadership and Governance | 5 | fair valu | intern control | statutori auditor | chief offic | vote right | commerci code | third parti | take account | real estat | joint ventur |

Having completed the first step, we proceed to re-doing the procedure for creating Figure 2 now instead for the 26 sub-dimensions. Figure 3 highlights, by circles, the positive correlations (grey < 0.5 and black > 0.5) between the sub-dimensions and the identified topics where *p*-values of less than 5% are indicated by an asterisk. There are seven distinct topics connected to GHG emissions, waste and hazardous materials management, ecological impacts, human rights and community relations, data security, supply chain management and competitive behavior. The other 19 topics are somewhat less distinctly connected to any particular sub-dimension. However, as an idealization we expected 26 distinct topics each pertaining to one of the 26 sub-dimensions. What is observed in Figure 3 is indistinct and we cannot readily corroborate or refute the claim of the intended users of the CSR reports to be shareholders. Following the testing strategy, we therefore move to step 3 and 4 as outlined in Table 1.

Under the claim, we expect shareholders to be the intended users of CSR disclosures irrespective of environment. However, we refute the claim if more stakeholder-oriented environments generate CSR reports non-aligned to the SASB dimensions. We therefore compare GRI or non-GRI compliant reports as well as reports of civil law and common law origin. Figure 4 gives the correlation matrix for GRI (a) and non-GRI (b). Figure 5 gives it for civil law (a) and common law (b). Both comparisons indicate a comparable concordance between the dichotomies and we therefore find the claim of the intended users of the CSR-reports to be shareholders to be corroborated. To formally test for comparable concordance, we employ permutation testing [79]. Under the presumption that, e.g., GRI and non-GRI, are comparable, the distribution of the chosen test statistic is obtained by simulations where the GRI/non-GRI indicator is randomly permuted across the reports. The observed test statistic is then compared to the distribution to approximate the *p*-value for a one-sided test. As test static we use the difference in the asterisks per group in Figures 4 and 5, respectively. For instance, and referring to Figure 4, the difference is −7 as there are 15 asterisks for GRI and 8 for non-GRI. Under the claim we would have expected less distinct correlations in the GRI group due to weaker alignment to SASB sub-dimensions and therefore fewer asterisks. On the contrary, the test statistic gives a negative value implying more distinct correlations in the GRI group. The approximate *p*-value equals 1 in testing comparable concordance. For the common law and civil law comparison the test statistic is −6 and the

approximate *p*-value again equals 1. To ensure that the result is not contingent on the rather ad-hoc choice of test statistic, we have also computed *p*-values for a test statistic that takes the difference in the number of black circles as well. Again, the *p*-value approximately equals 1 for GRI/non-GRI whereas it approximately equals 0.20 for civil law/common law. Hence, the third step implied that the claim was corroborated, and that finding was verified in the fourth step. Thus, we conclude that the intended users of the CSR reports are the shareholders.

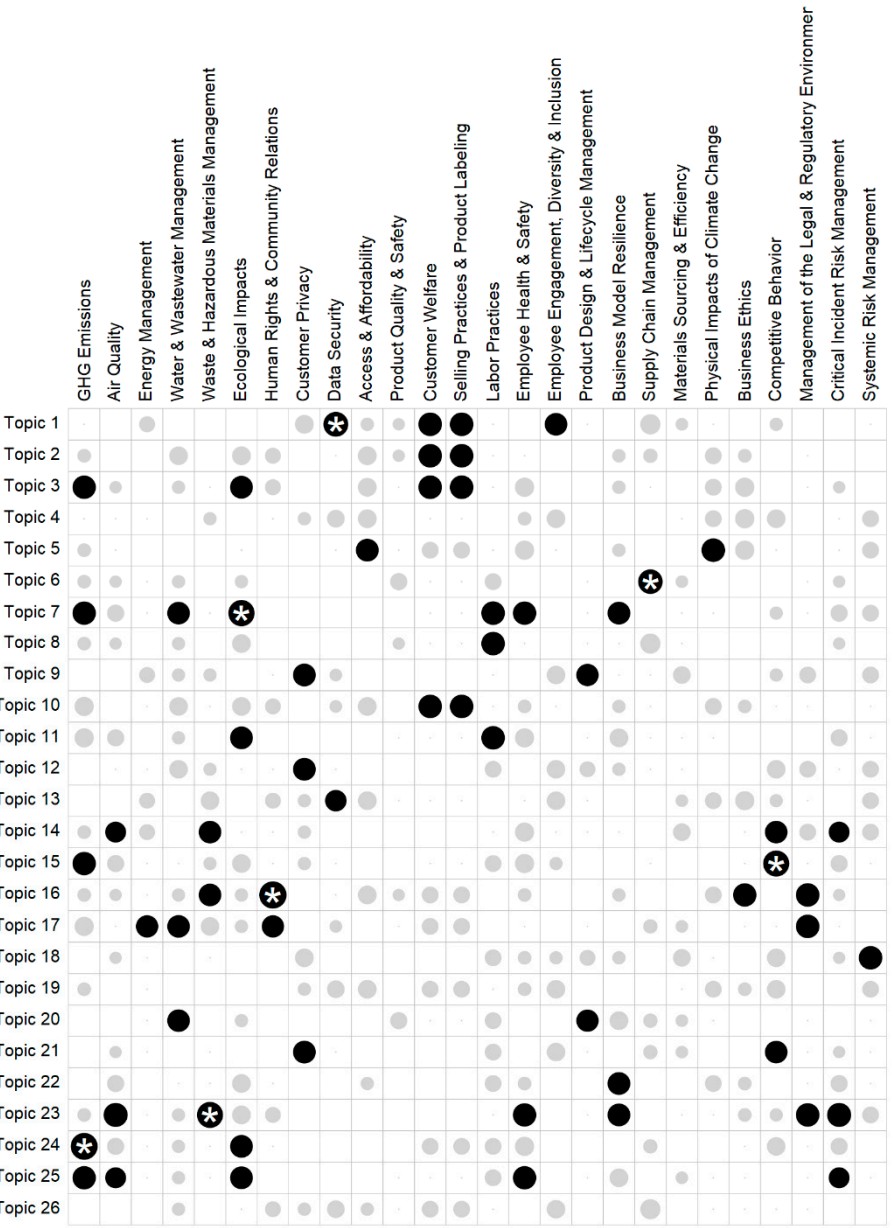

**Figure 3.** Correlation matrix for the 26 sub-dimensions and the data-driven topics with positive correlations (grey < 0.5 and black > 0.5) and where *p*-values of less than 5% are indicated by an asterisk (*).

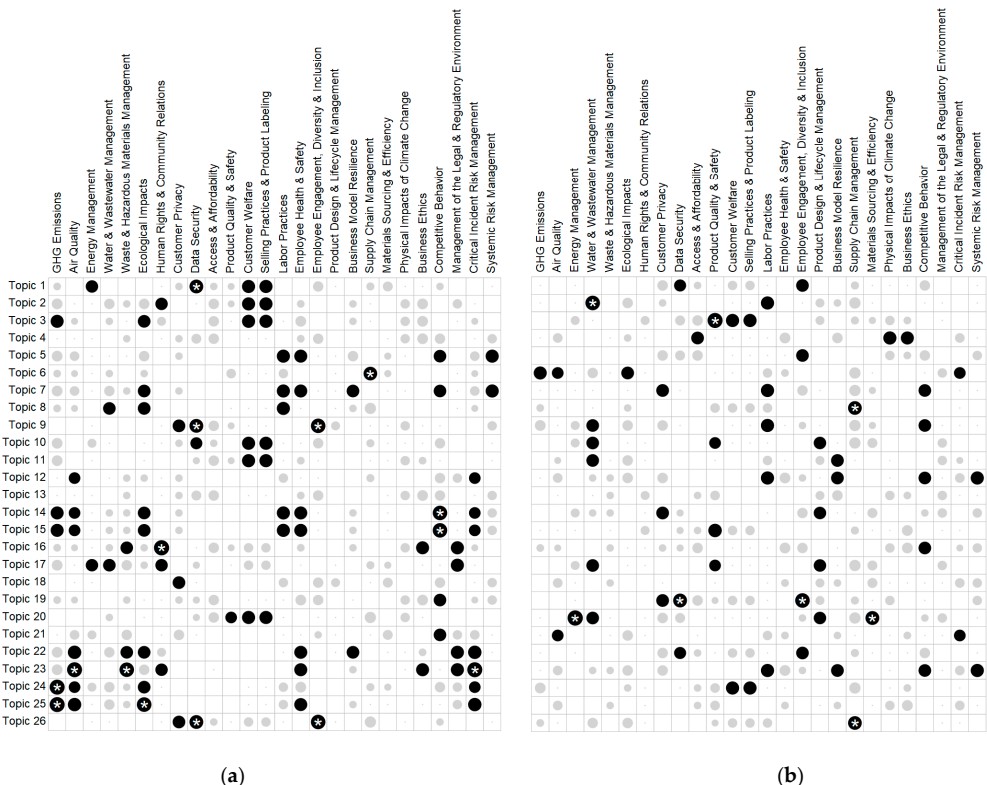

**Figure 4.** Correlation matrix for the 26 sub-dimensions and the data-driven topics with positive correlations (grey < 0.5 and black > 0.5) and where *p*-values of less than 5% are indicated by an asterisk (*) for (**a**) GRI compliant reports and (**b**) non-GRI compliant reports.

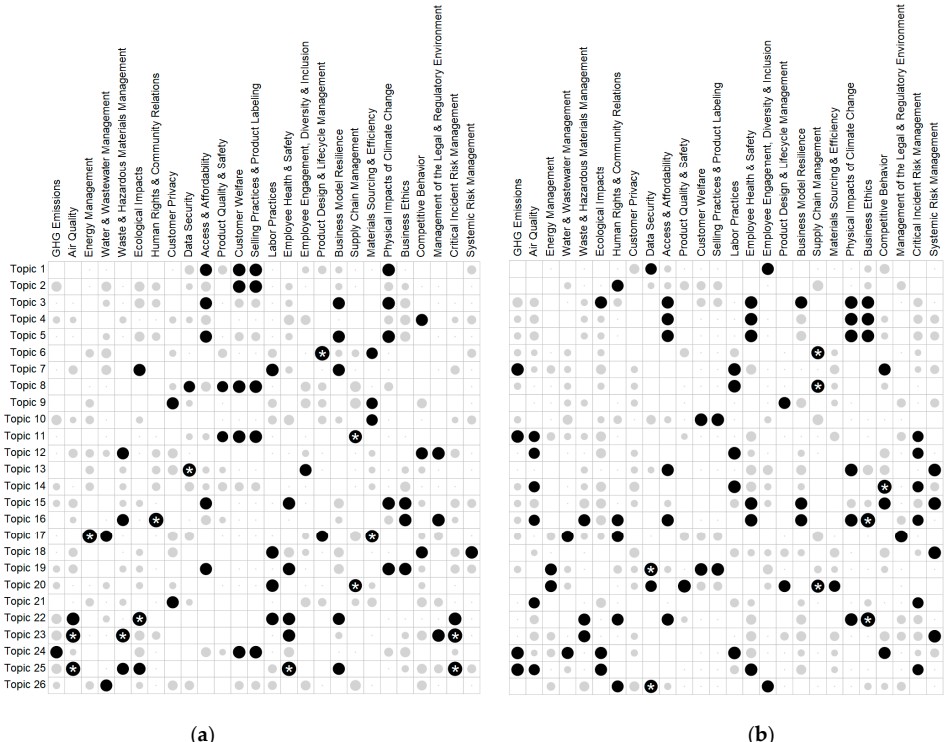

**Figure 5.** Correlation matrix for the 26 sub-dimensions and the data-driven topics with positive correlations (grey < 0.5 and black > 0.5) and where *p*-values of less than 5% are indicated by an asterisk (*) for (**a**) civil law reports and (**b**) common law reports.

## 5. Summary and Discussion

The evidence on materiality issues in non-financial disclosures such as CSR reporting and firms' intended users for these disclosures from a materiality perspective is limited. Furthermore, two fundamental yet unresolved questions in sustainability accounting are "(i) *who* needs to be accounted for, and (ii) *what* topics are relevant for corporate accounting." In this paper, we add evidence to these issues from a different perspective using a novel approach. We used the SASB materiality guideline of sustainability issues as a benchmark and investigated if firms' intended users of CSR disclosures are shareholders or stakeholders. The SASB standards by design helps us to discriminate between shareholders and stakeholders as material and immaterial issues in these standards are developed based on shareholders. We purported that firms' take shareholder perspective in CSR disclosure and found that firms in general do so. We found strong evidence when evaluated at the macro level of the five sustainability dimensions of the SASB and indication of consistency when evaluated at the micro level of the 26 sub-dimensions (i.e., the 26 general issue categories, as denoted by SASB). We also found that firms predominantly adopt a shareholder perspective in CSR disclosures even when using stakeholder-oriented reporting guidelines as well as in business environments, according to earlier literature, where stakeholders are favored. Lending to GRI's multi-stakeholder approach to CSR disclosures, we expected GRI reporters to adopt the wider stakeholder approach. However, we found this expectation to not hold up. Moreover, based on earlier literature, we expected firms operating in civil law legal environment to adopt stakeholder orientation while not so in the common law legal environment. Again, we did not find any clear indication as such, rather the results from our analysis pointed towards a shareholder orientation in both institutional environments.

The results are not surprising given SASB standards are developed based on historical value relevance—i.e., based on what firms found useful to disclose for shareholders. Nonetheless, alarmingly, since disclosures of GRI reporters also overlap substantially with SASB's reporting guidelines of material issue even though GRI guidelines take a multi-stakeholder approach and the process of determining materiality is thus fundamentally different. This raise concerns what scholars have long pointed out that firms are often mono-focused on shareholders [35–38], symbolic in dealing with stakeholders' concerns [80], and resort to green-washing or hypocrisy while undertaking CSR disclosures [6,8]. Even if we are to assume firms will sincerely align their value creation model towards stakeholders as proposed by Freeman [40], it can be extremely difficult for firms to report material CSR issues for multiple stakeholders in one single report in a decision-useful way. It is difficult not only because determination of materiality is complex, but also since multiple stakeholders may attach varying degree of importance to a particular CSR issue. Which is why perhaps research in materially aspect in both financial and nonfinancial accounting disclosures are extremely rare [24,55,56]. Nonetheless, to tackle this problem of materiality dilemma in CSR disclosures, some scholar has suggested issuance of multiple reports, generic vs. specialized reports (e.g., [81]), while others have suggested purposeful selection of stakeholders depending on core business activities while reporting on CSR dimensions (e.g., [23,36,37]). Both these approaches, however, can be costly, and requires synchronization of standard setters and regulators, and most likely regulatory intervention [23,82] and such type of interventions are costly for both firms and the society [83].

Nevertheless, these CSR disclosures, in their present form, are extremely difficult to process relying solely on manual investigation where a particular stakeholder may easily get overburdened due to sheer volume of disclosure, repetitions, and use of boilerplate language. To add to this complexity, the reporting practices vary greatly between firms, both within and across sectors. Our novel approach illustrates one possible way to deal with such types of complexity. We used an unsupervised Bayesian machine learning method to quantify the CSR reports' disclosures more objectively in a first step and triangulated the data-driven topics by statistical analysis and manual assessment–our approach helps to quantify issues that are reported in a substantial manner in these disclosures in a more

objective and effective way. This approach can aid decision makers to evaluate CSR disclosures as well as the effectiveness of reporting guidelines.

Finally, we provide direct evidence on how effective the voluntary reporting guidelines are (i.e., do they codify what firms disclose in their reports); in this case, we evaluated the SASB materiality framework for sustainability issues. Furthermore, most past studies that employ text analytics within the CSR disclosure domain are based on supervised learning method while we employ an unsupervised learning method—which is extensively data-driven and thereby minimizes the subjectivity in the analysis.

There are some limitations to note in our study. We chose to account for the year 2016 only, and the reason is two-fold. First, 2016 was the most recent year of complete data available on the GRI database at the time of collection and because G4, being the most recent guidelines, was only launched in 2013 thereby allowing some time for preparers to sync into the refined multi-stakeholder approach of GRI in these latest guidelines. Although not in the scope of this paper, multi-year analysis and additional data collection are ways to further explore the CSR reporting. While studying bigrams is useful for determining broader issues disclosed in these reports, it may also be that the context of sentences and sentiment analysis could be used to improve the topic identification and gauge whether an issue is expressed in a positive or negative way. Finally, since only reports written in English language were considered, some variation due to differences in cultural, norms, and style will be suppressed. Future studies may address some of these limitations.

**Author Contributions:** Conceptualization, A.M.H.; Data curation, C.L.; Formal analysis, C.L.; Investigation, C.L., A.M.H. and K.C.; Methodology, C.L. and K.C.; Supervision, K.C.; Visualization, C.L. and K.C.; Writing—original draft, C.L. and A.M.H.; Writing—review & editing, A.M.H. and K.C. All authors have read and agreed to the published version of the manuscript.

**Funding:** This research received no external funding.

**Institutional Review Board Statement:** Not applicable.

**Informed Consent Statement:** Not applicable.

**Data Availability Statement:** 3rd Party Data. Restrictions apply to the availability of these data. Data was obtained from GRI and are available https://database.globalreporting.org/ with the permission of GRI.

**Conflicts of Interest:** The authors declare no conflict of interest.

## Appendix A. Data Collection and Pre-Processing Procedures

The GRI database, which centrally holds sustainability disclosures of some 9500 firms around the globe has 47,093 aggregate data points for the years 1998 to the beginning of 2017 in multiple languages. Our first task was to extract the reports in PDF formats following the links provided in a excel file. To retrieve these reports (i.e., from firm websites) we used web scraping, and in the next step we used *n*-gram based language categorization to identify the reporting language.

We used the statistical software R for data collection, pre-processing, and analysis. For web scraping and conversion of PDF formatted files to text format purposes, we used the packages "downloader", "XML", and "RCurl". For language detection, we used the "textcat" package. It shall be noted here that out of 47,093 aggregate data points 36,192 reports had workable links to PDF documents on firm websites for the years 1998 to the beginning of 2017 and out of which 7881 reports were in English language. The reports include standalone sustainability or CSR reports, integrated reports, and annual reports that contains reporting on sustainability or CSR issues. We further choose to focus our analysis on reports for the year 2016, which is the final full year of our data, and omit GRI sectors conglomerates, services, renewable resource, and alternative energy, as well as others since there were less than 10 reports in each of these sectors. Note that the sector classifications for GRI and SASB are not identical and we recoded the sectors of GRI to SASB in the following manner, following SASB's classification: construction materials,

energy, metals products, and mining to 'extractives and minerals processing'; agriculture, food and beverage products, tobacco to 'food and beverage'; healthcare products and healthcare services to 'healthcare'; construction, energy utilities, railroad, real estate, waste management, and water utilities to 'infrastructure'; forest and paper products to 'renewable resources and alternative energy'; chemicals and equipment to 'resource transformation'; computers, media, technology hardware, and telecommunications to 'technology and communications'; and automotive, aviation, and logistics to 'transportation'. Finally, we also omit socialist legal origin reports since there were only 10 reports remaining in this particular legal origin (for instance, several sectors without any reports). Table A1 shows the final report count that were included in this study.

In the data pre-processing step, all the PDF reports were first converted to text files to remove all the formatting, those are then converted into matrices called document-term matrices. In a document-term matrix, the rows correspond to documents ($d$) and columns corresponds to terms or words ($w$). Thus, an element $m_{d,w}$ tells us how many times the $w$th term occurred in the $d$th document [84]. Next, the stop words are removed that are less meaningful [66]; these steps are part of what is known as the 'tidy' procedure [85]. One caveat of using term matrices is that it assumes the order of the words are not important (Blei et al. 2003), known as the 'exchangeability' assumption [86] which in the machine learning domain is known as the 'bag-of-words' assumption [1,84]. While the 'bag-of-words' assumption is a simplifying one, it does not jeopardize the performance of such models unless one requires to do some form of language modeling and the assumption only makes the computation more efficient [1]. This shortcoming, however, can be overcome using $n$-grams thereby including 'bigrams' (combination of two words) as opposed to 'unigrams' (single words) [87]. Although it has been found that higher $n$-grams does not lead to any significant improvement of the approximation results [69,88,89], we choose to allow for bigrams in our study. One outstanding concern is then 'sparsity', which is generally high when documents belong to different subjects [67]. In our case, all the documents are essentially CSR related disclosures, thus 'sparsity' is not of big concern. Lastly, the number of terms or words that appear in a document-term matrix can by reduced by either choosing a minimum number of occurrence or by a term-score [84]. We used term-frequency inverse document frequency (tf-idf) as the score as highlighted in Blei and Lafferty [90]. The score compares the count of the number of occurrences of each word in each document (i.e., term frequency) to the number of times each word occurs in the entire corpus (i.e., inverse document frequency). This results in a fixed-length document-term matrices instead of an arbitrary length. Furthermore, very little information is lost as the reduction is very small and the resulting document-term matrices with the words that have discriminative feature in the corpus [1]. This step is only used to reduce the word list, the analysis is based on the original term-frequency weighting calculated during the initial creation of the document-term matrices [84].

**Table A1.** Distribution of reports across sectors, SASB, and corresponding GRI sector label assignment for the year 2016

| SASB Sector Label | Report Counts | Corresponding GRI Sector Labels | Report Counts |
|---|---|---|---|
| Consumer Goods | 79 | Retailers | 37 |
| | | Households and Personal Products | 14 |
| | | Textiles and Apparel | 22 |
| | | Consumer Durables | 6 |
| | | Toys | 0 |
| Extractives and Minerals Processing | 139 | Mining | 38 |
| | | Energy | 67 |
| | | Construction Materials | 23 |
| | | Metals Products | 11 |
| Food and Beverage | 87 | Food and Beverage Products | 68 |
| | | Agriculture | 15 |
| | | Tobacco | 4 |

**Table A1.** *Cont.*

| SASB Sector Label | Report Counts | Corresponding GRI Sector Labels | Report Counts |
|---|---|---|---|
| Healthcare | 36 | Healthcare Products | 26 |
| | | Healthcare Services | 10 |
| Infrastructure | 119 | Energy Utilities | 25 |
| | | Real Estate | 45 |
| | | Construction | 33 |
| | | Water Utilities | 4 |
| | | Railroad | 8 |
| | | Waste Management | 4 |
| Resource Transformation | 51 | Chemicals | 29 |
| | | Equipment | 22 |
| Technology and Communications | 101 | Telecommunications | 40 |
| | | Media | 11 |
| | | Computers | 18 |
| | | Technology Hardware | 32 |
| Transportation | 59 | Logistics | 16 |
| | | Automotive | 22 |
| | | Aviation | 21 |

**Appendix B. SASB Standard Setting Process**

As of 2018, the SASB has published codified standards for 11 sectors (77 industries). These standards were developed from six years of research and extensive input from market participants. Initially, provisional standards for 10 sectors (79 industries) were published between July 2013–March 2016 to seek feedback on the decision-usefulness and cost-effectiveness of implementations of the standards from various stakeholders. Provisional standards were developed in the following major steps:

1. SASB staff conducted an evidence of materiality test of 43 generic sustainability issues for each industry. Which were then debated and reviewed by a Council and industry working groups (IWG) for preparation of "exposure draft of the provisional standards" for each industry.
2. The exposure drafts were then published for public review over a 90-day commenting period.
3. The public comments were then reviewed, responded, and incorporated as necessary, and the "industry research briefs" were prepared. The provisional briefs were also included with the published provisional standards.

This followed with a six-month consultation period between Q4 2016–Q1 2017 where the SASB staff gathered feedback from various stakeholders and prepared "technical agenda" and "basis of conclusion" for "exposure drafts of proposed changes to the provisional standards". The drafts were published for public review between 2 October 2017–31 January 2018. Finally, codified SASB standards for 11 sectors (77 industries) were published after considering the comments from the public on the exposure drafts. The codified drafts are monitored closely by SASB staffs for implementation and use of standards, as well as major trends and shifts in the market in the post-implementation review.

**Appendix C. Latent Dirichlet Allocation**

LDA is a three-level hierarchical Bayes model [1]. The LDA model in the first step draws a distribution of words ($w_1$, $w_2$, …, $w_n$) for a specified number of topics $T$, and this done at corpus level ($C$). The specified number of topics is not completely arbitrary rather depends on the judgement of the researcher given the underlying data (i.e., corpus). Moreover, there is a formal procedure, the perplexity score, to optimize the number of specified topics. In the second step, proportion of topics ($t_1$, $t_2$, …, $t_n$) is determined in

each document $(d_1, d_2, \ldots, d_c)$. In the third step a topic $t_n$ is chosen for each word $(w_n)$ in each document $(d_c)$ and finally given a topic $t_n$ the model choses the likely word generated in the first step.

The model involves drawing samples from Dirichlet distributions and multinomial distributions. Dirichlet distributions is a multivariate generalization of beta distribution while multinomial distribution is a generalization of binomial distribution. Furthermore, Dirichlet distributions is the conjugate prior of the multinomial distribution likelihood. There are two Dirichlet random variables in the LDA model, which are: (i) distribution of the specified number of topics in the corpus, and (ii) distribution of topics over the vocabulary. The multinomial distribution is the probability of two or more independent events, given the number of draws and fixed probabilities per outcome that sum to one. In this case, the outcomes are terms (i.e., words) and topics. For a more detail discussion on why this mixed-membership model is used please see [1].

More formally, as explained in Blei et al. [1] and Blei and Lafferty [90],

- A *word* is the basic unit of discrete data drawn from a vocabulary $(1, 2, \ldots, V)$. $V$ is the size of the vocabulary and is a $V$-dimensional Dirichlet with a vector parameter $\beta$, denoted by $Dir_V(\beta)$.
- A *document* is a sequence of $N$ words $(w_1, w_2, \ldots, w_n)$.
- A *topic* $(t_n)$ is a latent variable in each *document* while a group of *words*, determined through a stochastic process, defines each *topic*. The number of *topics* for the *corpus* is specified and follows a Dirichlet distribution, denoted by $Dir_T(\alpha)$, where $\alpha$ is a scaling parameter.
- A *corpus* is a collection of $C$ documents $(d_1, d_2, \ldots, d_c)$.

Finally, the probabilistic generative process is as follows

1. For each topic $t$, draws a distribution over words $P_t \sim Dir(\alpha)$
2. For each document $d$,
    a. Draws a vector of topic proportions $\theta_d \sim Dir(\beta)$
    b. For each word $w$,
        i. Draws a topic assignment $Z_{d,w} \sim Multinomial(\theta_d)$, $Z_{d,n} \in \{1, \ldots, T\}$
        ii. Draws a word $W_{d,w} \sim Multinomial\left(P_{Z_{d,w}}\right)$, $W_{d,w} \in \{1, \ldots, V\}$

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
