# Peer review of "Who Are the Intended Users of CSR Reports? Insights from a Data-Driven Approach"

_sustainability, doi:10.3390/su13031070_

Round 1
Reviewer 1 Report
Dear Authors,
Some points should be included within the manuscript in order to improve the publication.
1) Even though it is an interesting work and it is worthy of publishing, the practical applications of this study should be emphasized and analyzed in detail.
2) For readers to quickly catch the contribution in this work, it would be better to highlight major difficulties and challenges, and your original achievements to overcome them, in a clearer way in abstract and introduction.
3) It would be reasonable to rethink the “Summary and Discussion” section and to:
- highlight key findings in your “Results” section,
- place the paper within the context of how your research advances past research about the topic,
- describe how a previously identified gap in the literature (your literature review section) has been filled by your research,
- demonstrate the importance of your ideas and recommendations/suggestions,
- develop the limitations of your research,
- propose possible new or expanded ways of thinking about the research problem.
4) The paper should be reviewed by native English reviewers who will advise the authors about misprints and better sentence construction.
I do hope you find the comments helpful as you move forward with your paper.
Author Response
Response covers all three reviewers. Please see the attachment.

Reviewer 2 Report
Dear authors thank you for the opportunity to review your paper. I think the chosen topic is quite interesting, however, at present the paper has some weaknesses.
Firstly, the introduction should better explain the research field your paper fits into and should make the research question more engaging.
Secondly, the paper should include a reference theory (probably a solution could be to combine shareholder theory and stakeholder theory).
I would like to point out some recent papers that could be interesting and useful for the theoretical part:
- Hörisch, J., Schaltegger, S., & Freeman, R. E. (2020). Integrating stakeholder theory and sustainability accounting: A conceptual synthesis. Journal of Cleaner Production, 275, 124097.
- Vitolla, F., Raimo, N., Rubino, M., & Garzoni, A. (2019). The impact of national culture on integrated reporting quality. A stakeholder theory approach. Business Strategy and the Environment, 28 (8), 1558-1571.
- Vitolla, F., Raimo, N., Rubino, M., & Garzoni, A. (2019). How pressure from stakeholders affects integrated reporting quality. Corporate Social Responsibility and Environmental Management, 26 (6), 1591-1606.
- Thijssens, T., Bollen, L., & Hassink, H. (2015). Secondary stakeholder influence on CSR disclosure: An application of stakeholder salience theory. Journal of Business Ethics, 132 (4), 873-891.
It is also necessary to insert a discussion section and a conclusions section. In the discussion section the authors should discuss the results obtained by anchoring them to the reference theory. In the conclusions section instead they should summarize the results and then insert the managerial implications, limitations and ideas for future research.
Good luck.
Author Response

(The authors gave the same response as above.)

Reviewer 3 Report
I find this paper quite interesting and important for science but also for the professional public. The paper is well-written and brings new insights into the area of CSR disclosures and reporting.
Although I find this paper quite interesting, I would like to suggest a few potential areas for improvement:
- Please, make the theoretical and practical implications of the paper more visible and understandable.
- It would be good to try to make a comparison between your results and the results of some similar studies on the same theme.
- Please, consider the following sources to include in your paper:
-
- Watts, G., Fernie, S., & Dainty, A. (2019). Paradox and legitimacy in construction: how CSR reports restrict CSR practice. International Journal of Building Pathology and Adaptation.
- Friske, W., Nikolov, A. N., & Hoang, P. (2020). CSR reporting practices: an integrative model and analysis. Journal of Marketing Theory and Practice, 28(2), 138-155.
- Grubor, A., Berber, N., Aleksić, M., & Bjekić, R. (2020). The influence of corporate social responsibility on organizational performance: A research in AP Vojvodina. Anali Ekonomskog fakulteta u Subotici, (43), 3-13.
- Aggarwal, P., & Singh, A. K. (2019). CSR and sustainability reporting practices in India: an in-depth content analysis of top-listed companies. Social Responsibility Journal.
Author Response

(The authors gave the same response as above.)

Round 2
Reviewer 2 Report
Well done, congratulations to the authors.
Author Response
Thank you very much for your review.